# Reproducibility report for "Interpretable Complex-Valued Neural Networks for Privacy Protection"

## Reproducibility Summary

**Scope of Reproducibility**

The original work by Xiang et al. (2020) claimed (1) that complex-valued DNNs effectively increase the difficulty of inferring inputs for the adversary attacks compared to the baseline. In addition, Xiang et al. (2020) stated that the (2) proposed privacy-protecting complex-valued DNN effectively preserves the accuracy when compared to the baseline.

**Methodology**

Since the original paper's code was not published, all of the codebase was written independently from scratch, based solemnly on how it was described in the paper. We mostly used a Nvidia's *RTX 2060 Super* as the GPU and a *AMD Ryzen 3600x* as the CPU. The runtime of each model was highly dependant on the architecture used. The runtimes for each model can be found in Table 2.

**Results**

In contrast to the first claim, we have discovered that for most of the architectures, reconstruction errors for the attacks are quite low, which means that in our models the first claim is not supported. We also found that for most of the models, the classification error is somewhat higher than those provided in the paper. However, these indeed relate to the original work and partially support the second claim of the authors.

**What was easy**

Authors of the original paper utilized famous architectures for some of architectures' parts, such as ResNet and LeNet, that were well explained and defined in the literature. In addition, authors, provided formulas on the modified rotation-invariant Complex DNN modules (ReLU, max pooling etc.), implementation of which was relatively straightforward. The paper was based on the openly available datasets.

**What was difficult**

The paper did not provide any information on the architecture of the critic for the WGAN, along with the architecture of the angle discriminator utilized in inversion attack 1. It also does not provide any information about crucial hyperparameters, such as the k value used for k-anonimity.

**Communication with original authors**

We did not contact the original authors of the publication.

# 1 Introduction

Deep Neural Networks (DNNs) can process a massive data volume but require great computing power to process this data. Therefore it is an interesting option for small devices like smartphones or IoT devices to use a cloud operator for these computationally expensive tasks. Although this is an efficient way to process data, the cloud operator is susceptible to privacy threats. A potential attacker could reconstruct or infer private properties of the data. Possible solutions are subject of current research.

Xiang et al. (2020) proposed a possible solution using encryption and complex-valued neural networks to address this problem. They showed that their approach increases the difficulty of inferring inputs or properties from intermediate layer features. Our paper aims at reproducing their findings.

Xiang et al. (2020) extended the standard DNN by encrypting the intermediate layer features using complex-valued DNNs (Trabelsi et al. (2017)). The local device encodes the original input into intermediate features. Those features are encrypted and sent to a processing unit, a complex-valued neural net located in the cloud. It does the computationally expensive operations while not being able to infer properties of its input. The result is sent back to the local device and decrypted.

Ideally, the local device is able to decrypt the encoded data using the secret key. However, an adversary shouldn't be able to decrypt the features without the key. Xiang et al. (2020) achieved this by rotating intermediate layer features into the complex plane using a random angle. The angle acts as a key and can be used to reverse the rotation. The processing unit consists of rotation-invariant operations only. Thus, the local device can reverse the rotation after receiving the results from the processing unit. To make it hard to deduce the angle, a generative adversarial network is trained to alter the intermediate features to introduce obfuscation whilst keeping important information.

# 2 Scope of reproducibility

The main problem Xiang et al. (2020) addresses is the danger of adversary attackers being able to recover original inputs or hidden properties of the input. Xiang et al. (2020) claim their complex-valued model to be robust against these kinds of adversary attacks and show this by attacking their model with various inversion and inference attacks. Here, they claim that attacking their complex-valued DNNs acquires greater reconstruction loss than when they attack the original DNNs, meaning they are more resistant to adversary attacks. Furthermore, even though the input is encoded, they claim that their complex-valued DNNs have almost the same utility performance (classification error rates) as the original DNNs.

In this paper, we test the following concrete claims:

1. Xiang et al. (2020) proposed complex-valued DNNs effectively boost the difficulty of inferring inputs for the adversary compared to the baseline.

2. Xiang et al. (2020) proposed their privacy-preserving complex-valued DNN largely preserves the accuracy when compared to the baseline.

# 3 Methodology

The paper from Xiang et al. (2020) was replicated [1] by solely using the methods described in their paper, since there was no implementation available online. Our approach involved implementing the models used by Xiang et al. (2020), which included the ResNet-$\alpha, \beta$-20/32/44/56/110 (He et al. (2016)), the LeNet (Lecun et al. (1998)), the VGG-16 (Simonyan and Zisserman (2015)) and the AlexNet (Krizhevsky et al. (2012a)). Most of these papers had implementations available online, which were adjusted slightly to be able to process complex-valued intermediate layer features. Furthermore, these DNNs had to be disassembled to create the complex-valued DNN structure proposed by Xiang et al. (2020) that consists of an encoder, processing module and decoder.

The encoder is in principle the same as the first few layers of the used DNN, but at the end of the encoder a complex rotation is applied (Eq. 1).

$$x = \exp(i\theta)[a + bi] \tag{1}$$

---

[1] Our implementation can be found at our GitHub (Authors (2021))

The value a represents the regular input that is put through the encoder. The value b is the fooling counterpart that can be seen as a different input that is also put through the encoder. Theta is picked to be random angle, which will act as the secret key we mentioned earlier.

The resulting features are then fed to the processing unit. The processing unit consists of the middle layers of the used DNN. The processing unit has to deal with complex features, which means adjustments had to be made to the regular functions that the DNN uses. We used the proposed functions in Trabelsi et al. (2017) to achieve this. Furthermore it is important that the processing unit does not change the rotation of the features, because otherwise if we try to rotate it back with our randomly picked angle we will get random results. Therefore we also had to make the used functions rotation invariant. Methods for achieving this were described in Xiang et al. (2020).

Finally the features are put through the decoder, which consists of the final layers of the used DNN. The decoder first rotates the features back with the randomly picked angle (Eq. 2) and then feeds the result to the final layers.

$$\hat{y} = d(R[h \exp(-i\theta)]) \tag{2}$$

The value h represents the output of the processing unit and the value $\hat{y}$ is the prediction that the decoder d made.

Xiang et al. (2020) also uses a Wasserstein Generative Adversarial Network which was proposed by Arjovsky et al. (2017) and is part of the encoder. This WGAN consists of a generator and a critic that use complex rotations to teach the network to generate synthesized features to hide the original ones. Since no implementation with a WGAN that works with complex-valued features was available online, this WGAN encoder had to be completely remade.

Lastly, for the adversary attacks, Xiang et al. (2020) attacked the model with inversion and inference attacks. For the inversion attacks, we adopted the U-net model proposed by Ronneberger et al. (2015), which had similar variants online that had to be hardly modified to fit their implementation. For the inference attack, a model had to be implemented that functions as a classifier to predict hidden properties of the input.

## 3.1 Model descriptions

### 3.1.1 Complex-valued DNNs

In Xiang et al. (2020) approach, the DNN is split into two local parts that are used to encode and decode the data (encoder and decoder, respectively) and a middle part that performs all of the heavy data processing (processing module).

In the original work, two ResNet implementations were described: ResNet-$\alpha$ and ResNet-$\beta$. In ResNet-$\alpha$ the input is transformed until the first 16x16 feature map, from where the output is sent to the encoder. After the encoding, the processing module transforms the data until the first 8x8 feature map. From that point on, all following layers constitute to the decoder. The difference in ResNet-$\beta$ is that the decoder was composed by the last residual block and the layers following it.

For describing the other classical DNNs, we only specify the encoder and the decoder, where all the remaining layers contribute to the processing unit. For the LeNet model, the encoder consisted of the first convolutional layer and the WGAN, whereas the decoder contained only the softmax layer. In the VGG-16 all layers before the last 56x56 feature map constituted the encoder. Here, the decoder consisted of fully-connected layers and the softmax layer. For the AlexNet, the first three convolutional layers' output was fed into the encoder, where the decoder contained fully-connected layers and the softmax layer.

### 3.1.2 WGAN

To introduce obfuscation and make it hard for the adversary to reconstruct the original features, a WGAN is utilized. A WGAN is a variant of a generative adversarial network known to be more resistant against hyperparameters or mode collapse compared to the original approach. The WGANs encoder is used to introduce obfuscation to the features. The critic is only used to train the generator, and its objective is to distinguish rotated features from those without rotation. The generator of the WGAN shares the same network as the encoder we described earlier. This means that one part of the network (the encoder/generator) is trained with two different purposes; One is for classification and the other is for the WGAN. To train the generator, the encoded and rotated (Eq. 1) features it produced are passed to the critic, which is trying to retrieve the original features by rotating the given features back.

$$a' = \Re[x \exp(-i\theta')] \tag{3}$$

The critic creates k-1 fake samples and rotates them by k-1 randomly sampled angles $\theta'$ (Eq. 3), where $\theta' \neq \theta$. The critic then discriminates whether these rotated features are close to the original complex-valued features.

Table 1: Relevant statistics for the used datasets

| Dataset | Labels | Number of examples | Split (train/dev/test) |
|---------|--------|--------------------|------------------------|
| CIFAR-10 | 10 | 60.000 | 50.000 / 0 / 10.000 |
| CIFAR-100 | 100 | 600.000 | 500.000 / 0 / 100.000 |

From Xiang et al. (2020), it was unclear whether the generator is only a part of the encoder or whether the WGAN trains all encoder layers. Because no architecture was given for the generator, we decided to train all the encoder layers with the WGAN loss and not introduce a stand-alone generator after the encoder.

The original WGAN uses weight clipping for the critic network. However we found that all the weights converged to the clipping values rather quickly. Because of this we cannot train the WGAN appropriately. To fix this issue we used a gradient penalty which was introduced in Gulrajani et al. (2017).

### 3.1.3 Adversary models

**Inversion attacks**
The objective of the inversion attack was to reconstruct the input images from the encoded intermediate features. Xiang et al. (2020) implemented two inversion attacks: in inversion attack 1 a new discriminator ($D'$) is first trained to predict the most probable features ($a^*$) by learning the most likely angle at which the intermediate layer features are rotated (Eq. 4). The most probable features (Eq. 5) are then used to help train a decoder model that tries to reconstruct the original images (Eq. 6). In inversion attack 2 the angle prediction discriminator is not included and the attacker only trains a decoder that tries to reconstruct the original images from the given intermediate layer features.

$$\hat{\theta} = \max_{\theta} D'(\Re[x \exp(-i\hat{\theta})) \tag{4}$$

$$a^* = \Re[x \exp(-i\hat{\theta})] \tag{5}$$

$$\hat{I} = \dec(a^*) \tag{6}$$

The structure of the reconstruction model for both inversion attacks was based on a modified U-Net (Ronneberger et al. (2015)) described in the original work. U-Net is a neural network architecture widely used for image segmentation. It is based on an Autoencoder architecture decorated by skip connections. The skip connections help to reconstruct the exact low-level attributes such as the location of edges. Intermediate layer features from the encoder are scaled up to the original input size and then fed into the U-Net, aiming to reconstruct the original image.

**Inference attacks**
During inference attacks a classifier is trained on either similar raw images (inference 1), rotated features $a^*$ (inference 2) or fully reconstructed images $\hat{I}$ (inference 3). Furthermore, a model is trained using k-nearest neighbors (k-NNs), where the attacker compares $a^*$ against features of each training example to find similarities in the training set. We did not implement the inference attacks, because the results of the inversion attacks showed that the privacy protection was not working properly. Instead of implementing the inference attacks, which would yield similar poor results, we decided to investigate further into why the privacy protection was not working.

### 3.2 Datasets

Xiang et al. (2020) used CIFAR-10/100 (Krizhevsky et al. (a), Krizhevsky et al. (b)) to train the ResNets and LeNet, CUB-200 (Welinder et al. (2010)) to train VGG-16 and CelebA (Liu et al. (2015)) for AlexNet. Since we did not implement VGG-16 and AlexNet, we only used the CIFAR-10/100 datasets using the train/test split described in Table 1. Each split was halved to create two smaller datasets used for training/testing either the privacymodel or the adversary attacker. Before training, all datasets were normalized.

### 3.3 Hyperparameters

Unfortunately, none of the standard hyperparameters such as learning rate, optimizer, weight decay, etc. were mentioned in the paper. Therefore we had to adapt and choose them ourselves. For the WGAN, we used the hyperparameters given in the original implementation by Arjovsky et al. (2017). We implemented weight clipping and gradient penalty. For the first one, we used RMSprop with a learning rate of $5e^{-5}$ for both the generator and the critic. Weights were clipped between $-0.01$ and $0.01$ for the Critic's weights.

As this learning rate was too low for the classification task, we used a different optimizer. Adam was used with a learning rate of $5e^{-4}$. For the gradient penalty approach, we set lambda to $10$ and used Adam with a learning rate of $5e^{-4}$. We did not do an intensive hyperparameter search to optimise these parameters.

Finally, another hyperparameter specific to this paper is called k. k represents the number of times the discriminator decides on an input per iteration. The discriminator always has to calculate a real score based on the real features a, so there are k-1 fake inputs that determine the fake score. This hyperparameter is also never defined in the paper. We set the k value on 8, and we have not been able to test other options, unfortunately.

We trained our adversary networks with the Adam optimizer and a learning rate of $5e^{-4}$. For training the U-net we used the Mean Squared Error loss and for training the angle predictor network we used the Absolute Mean loss.

### 3.4    Experimental setup and code

Since PyTorch currently does not fully support complex-valued tensors, we chose to split up the 'real' part and the 'imaginary part' into two tensors, where we have created new complex functions to process these two tensors correctly.

We introduced complex and, most-importantly, rotation-invariant ReLU, BatchNorm, and MaxPool layers according to the original work's formulae. In addition, we have also discovered that for Complex Linear Layers, the bias term should not be involved in matrix computations, even though it was only mentioned for the case Complex Convolutional Layers in the original paper.

The architecture of processing unit is designed so that after the the features $x$ from encoder are passed through processing unit, we could express the output as a complex rotation of outputs of processing unit, i.e $\Phi(x)$, specifically it means:

$$\Phi(e^{i\theta}x) = e^{i\theta}\Phi(x) \tag{7}$$

To keep the equation true for the Convolutional Linear Layers $\Phi(e^{i\theta}x)$ and $e^{i\theta}\Phi(x)$ should be equal to each other, meaning that for:

$$\Phi(e^{i\theta}x) = w \cdot (e^{i\theta}x) + b \tag{8}$$

$$e^{i\theta}\Phi(x) = e^{i\theta}(w \cdot x + b) \tag{9}$$

Given any $\theta$, we see that the equality would hold only if b = 0, which would mean that no bias term would be needed. Thi is exactly the objective of rotation invariance for our network as it was mentioned in the publication.

In the original work, the Critic's architecture was not described, so some assumptions regarding its architecture were made. We found that one linear layer is not sufficient. Thus, we added a convolution, a ResNet block and another convolution. Since the LeNet network looks very different from the ResNet networks we decided to change to the critic architecture to look more like the generator of LeNet.

In the paper, they describe how the decoder of the AlexNet only consists of a softmax layer. This is not possible, because they train the network on the attributes of CelebA and since multiple attributes per image are used, introduction of a sigmoid function is essential. The output of the last layer in the decoder is a sigmoid and the binary cross-entropy loss is applied in order to estimate how well it assigns attributes to each image.

The paper implements inversion and inference attacks to test whether it is preserving the privacy. For inversion attack 1 it uses a network to determine the angle that was most likely used for rotation. We decided to implement this network by creating a network that is identical to the critic that we used for that specific network.

The U-Net architecture used in the inversion attacks is constructed using 6 convolutional layers per block, instead of the standard 2 convolutional layers per block. Furthermore, the U-net architecture consisted of 4 down and 4 upsampling blocks. Each downsampling block reduced the features' dimensions by 2, while the up-sampling ones doubled the aforementioned dimensions. Thus, input and output image widths and heights were preserved.

We found that the inversion attacks were reconstructing the images very well when we used them on our trained networks. Adding a convolution layer without a ReLU activation function to our generator increased the overall reconstruction loss, effectively making our network preserve the privacy better. This was further improved by randomly swapping a and b in our generator, which lead to a significant increase in the reconstruction loss.

From all the networks that were used in Xiang et al. (2020) we implemented LeNet and the ResNet-32/44/56/110-$\alpha, \beta$ architectures. We did not implement the other architectures, because the ResNet and LeNet architectures showed that

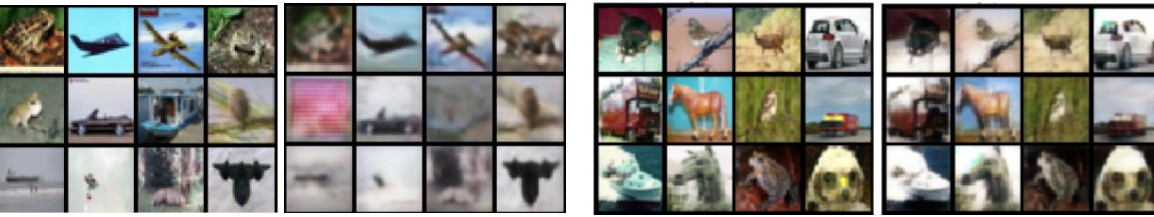

LeNet-5 architecture                                ResNet-32-$\alpha$ architecture

Figure 1: Image reconstruction after inversion attack 1. Original images on the left,
reconstructed images on the right of each atchitecture

the privacy protection was not working as described. We decided to continue our investigation into ResNet and LeNet, instead of implementing more networks which would face the same problems as ResNet and LeNet.

### 3.5 Computational requirements

We mostly used a RTX 2060 super for the GPU and a ryzen 3600x for the CPU. The runtime of each model was highly dependant on the architecture used. The runtimes for each model can be found in Table 2.

<table>
<tr><td colspan="2">Table 2: Average runtime for each model</td></tr>
<tr><td>Model</td><td>Average Runtime</td></tr>
<tr><td>LeNet</td><td>48 m</td></tr>
<tr><td>ResNet-32-$a$</td><td>2h 29m</td></tr>
<tr><td>ResNet-32-$b$</td><td>2h 41m</td></tr>
<tr><td>ResNet-44-$a$</td><td>3h 15m</td></tr>
<tr><td>ResNet-44-$b$</td><td>3h 08m</td></tr>
<tr><td>ResNet-56-$a$</td><td>3h 34m</td></tr>
<tr><td>ResNet-56-$b$</td><td>3h 23m</td></tr>
<tr><td>ResNet-110-$a$</td><td>3h 20m</td></tr>
<tr><td>ResNet-110-$b$</td><td>3h 12m</td></tr>
</table>

Table 3: Inversion attack 1 and 2 results

| Model | Inversion Attack 1 | | Inversion Attack 2 | |
| --- | --- | --- | --- | --- |
| | **Paper** | **Reproduced** | **Paper** | **Reproduced** |
| LeNet | 0.2405 | 0.1499 | 0.1027 | 0.1244 |
| ResNet-32-$\alpha$ | 0.2569 | 0.0277 | 0.2412 | 0.0464 |
| ResNet-32-$\beta$ | 0.2515 | 0.0292 | 0.2425 | 0.0323 |
| ResNet-44-$\alpha$ | 0.2746 | 0.0256 | 0.2419 | 0.0293 |
| ResNet-44-$\beta$ | 0.2511 | 0.0190 | 0.2397 | 0.0383 |
| ResNet-56-$\alpha$ | 0.2804 | 0.1031 | 0.2377 | 0.0399 |
| ResNet-56-$\beta$ | 0.2585 | 0.0242 | 0.2358 | 0.0483 |
| ResNet-110-$\alpha$ | 0.3081 | 0.0292 | 0.2495 | 0.0447 |
| ResNet-110-$\beta$ | 0.2582 | 0.0207 | 0.2414 | 0.0321 |

## 4 Results

### 4.1 Results reproducing original paper

In these sections we show the results produced by our network and relate them to the claims introduced in section 2.

#### 4.1.1 Increased difficulty of inferring inputs

In this section we show the results that relate to the first claim (1) based on what our initial implementation of our networks produced. Table 3 shows the reconstruction errors of multiple different networks with the old implementation when attacked with inversion attack 1 and 2. The table shows that the reconstruction errors of our networks are much lower than the reconstruction errors of the paper's networks. We can therefore say that the first claim is not supported by these networks. The reconstructed images that were created by inversion attack 1 on the ResNet-32-$\alpha$ network can be seen in Figure 1b. The reconstructed images look very similar to the original images, further proving that the first claim is not supported.

In Table 3 we can see the results of attacking our new implementation network with inversion attack 1 and 2. Here we see that the results look a lot more similar and we can therefore say that the first claim is supported under our new implementation. This claim is further supported by Figure 1 where the reconstructed images can be seen of the new implementation network.

#### 4.1.2 Preservation of accuracy

In this section we show the results that relate to the second claim (2). Table 4 shows the classification errors of the old implementation of our networks. The table shows that our network's classification errors are quite a bit higher than the

Table 4: Classification Errors of the paper's results (left) and the reproduced results (right)

| | Classification Error | |
|---|---|---|
| **Model** | **Paper** | **Reproduced** |
| LeNet | 17.95 | 59.62 |
| ResNet-32-$\alpha$ | 10.48 | 19.53 |
| ResNet-32-$\beta$ | 11.12 | 25.00 |
| ResNet-44-$\alpha$ | 11.08 | 26.09 |
| ResNet-44-$\beta$ | 10.51 | 26.48 |
| ResNet-56-$\alpha$ | 11.53 | 25.78 |
| ResNet-56-$\beta$ | 11.28 | 28.91 |
| ResNet-110-$\alpha$ | 11.97 | 24.22 |
| ResNet-110-$\beta$ | 11.85 | 28.17 |

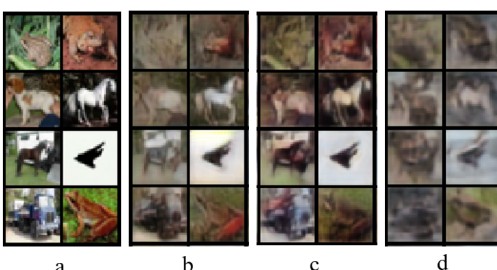

a  b  c  d

Figure 2: Image reconstructions, trained on Cifar10 and ResNet-22-$\alpha$. a) Original images. b) Model without ReLU at the end of the generator. c) Model with ReLU at the end of the generator. d) Model trained while randomly swapping a and b; The WGAN was not trained, however, the adversary is not able to reconstruct the inputs. Also, the accuracy of the classifier didn't change using this approach.

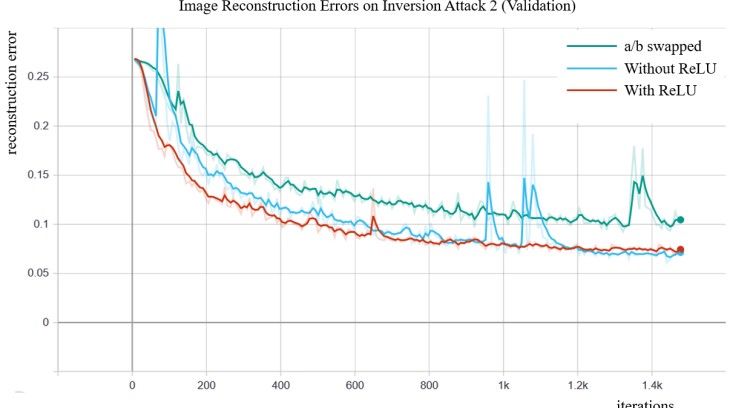

Figure 3: Training of the adversary model. Random swapping of a and b yields a significantly higher reconstruction error. The models with and without ReLU have similar reconstruction errors, but the model without ReLU needs much longer to train as it introduces better obfuscation.

paper's classification errors. However our network's classification errors are still quite low, which is why the results partially support the second claim. In table 4 we can see the classification errors of the new implementation of our networks. We see that the classification error is quite similar to the old implementation and therefore also partially supports the second claim.

## 4.2   Results beyond original paper

### 4.2.1   Linear activation at the end of the generator

The encoded features are rotated in the complex plane. Thus, their values can become negative. However, if a and b contain only positive values due to a ReLU activation at the end of the WGANs generator, finding the initial angle would be easy for an adversary. Therefore, we concluded that the ReLU has to be replaced or omitted. Empirically, we confirmed this as we found that the model spreads obfuscation and the adversary needs much more time to train. However, when training until convergence, we weren't able to confirm a significant difference in reconstruction error when compared to the model with ReLU. Reconstructed images are compared in Figure 2a-c. This underlines our general concern regarding the WGAN being too weak compared to the adversary. Again, we want to pay attention to the adversary's slow convergence rate when not using the ReLU activation. At first glance, the model seems to converge with a high reconstruction error which might lead to a wrong conclusion if training is stopped too early.

#### 4.2.2 K-anonymity is questionable

There is no good way to test how the GAN is performing except training an adversary. If the adversary is trained poorly, there could be another adversary that is able to reconstruct the images much better. To test the adversary, we disabled the training of the WGAN and instead randomly swapped the encoded features $a$ with the randomly sampled feature $b$. The decoder had access to this information and thus was able to perform as good as the normal classifier. Surprisingly, the adversary performed really poor and was not able to reconstruct the images. Instead, the adversary reconstructed an image that was very blurred and an interpolation between the two images. Our results are shown in Figure 2.

Comparing the reconstructed images to the images of Xiang et al. (2020), we found that they look really similar. This skepticism is further supported by the fact that the images shown in appendix B of Xiang et al. (2020) seem to be an interpolation of two images. Finally the average angle errors shown in table 4 of Xiang et al. (2020) are all $\pi/4$ except for one (VGG-16). This seems questionable since this is equal to the initial angle that lies exactly in between a and b. These reasons combined make us doubt the k-anonimity that is claimed and makes us contemplate that it is 2-anonimity instead. This 2-anonimity means that the results are just interpolations of a and b, and the attacker just has to deceiver on from the other.

All of this evidence suggests that an adversary could be able to reconstruct two images if it would be designed to do so. However, this questions the k-anonimity, which could be reduced to a 2-anonimity by selecting a better adversary.

## 5 Discussion

From the results of the initial implementation of the networks (without removing the relu and adding the swapping of a and b in the generator) we can see that we do not reproduce the results of Xiang et al. (2020). Our networks don't protect the privacy as the images can be perfectly reconstructed and the properties can be perfectly inferred. Therefore the first claim (1) is not supported by our results. The classification errors are however quite low and even though they are higher than those from Xiang et al. (2020), we think that with the right parameters we could have achieved the same classification errors as them. Therefore we do think that the second claim (2) is supported even though our results only show partial support.

The splitting of the networks in Xiang et al. (2020) is questionable. The whole idea of the paper is to have a processing unit running on the cloud so small IoT devices do not have to deal with the computational load themselves. This of course introduces privacy concerns, which the paper aims to address. However when we look at the way Xiang et al. (2020) splits the networks into three parts, we see that a lot of computational effort is being put on the encoder and decoder networks. When we look at how they split the ResNet networks for example, approximately only one third of the computational effort is put on the processing unit. If this way of splitting is the only way the privacy can be protected, then it is questionable whether networks like this can actually be used on IoT devices. Also, the local device has to keep the dataset during inference in order to sample $b$. However, IoT devices lack storage capacity which could lead to major difficulties in practice.

The way Xiang et al. (2020) show their results of the inference attacks is questionable as well. They decided to evaluate the classification of the network performing the attack and the evaluation of the privacy of their network on different parts of the datasets. From an evaluation point of view, we do not understand why they did this. What is worse is that it seems they chose the dataset parts in such a way that their results seem better than they actually are. For example, they evaluate the classification on the 20 major super classes of CIFAR-100, but they evaluate the privacy on all 100 classes. By definition this means that the privacy error will be higher than the classification error, which is exactly what they want, hence why it seems a bit like cheating.

The choice of the U-Net as the adversary is unclear to us. We don't see any value in first upsampling the features followed by downsampling them in the first part of the U-Net. In addition, the input of the U-Net has no important low-level features as they are upscaled, so the value of the skip connections is questionable to us. We think it'd be sufficient if the adversary consists of a simple decoder that reconstructs images by upsampling and applying convolutions.

### 5.1 What was easy

Overall we found Xiang et al. (2020) quite difficult to understand and reproduce, but what did make it a lot easier was the different implementations that were already online. For most networks that they used we found an implementation online that we could use and especially the complex neural network implementations were very useful.

- Datasets, such as CIFAR-10 (Krizhevsky et al. (a)), CIFAR-100 (Krizhevsky et al. (b), CelebA (Liu et al. (2015)) are publicly available and are easy to get access to.

- Main architecture for the privacy model is based on the existing and well-documented architectures such as ResNet (He et al. (2016)), LeNet(Lecun et al. (1998)), AlexNet(Krizhevsky et al. (2012b)) etc.

## 5.2 What was difficult

Some crucial details such as discriminator architecture and major hyper-parameters were not included in the publication, making it difficult to properly replicate the results with sufficient precision. One of the main hyper-parameters that were missing were the learning rates, optimizers and weight decay values that were used. This particularly, made implementation of WGAN vague and we had to make a lot of assumptions. This is especially a problem, since the WGAN is the only part that can provide the privacy protection, so if the WGAN does not work properly we cannot achieve the privacy protection that they claim in the paper.

Description of the inversion attacks (1) and (2) mentioned in section 4.3 were not consistent with the description of experiments in section 5. For example, the authors state in the section 4.3 of the paper that inversion attack 1 implies estimation of proper angle $\hat{\theta}$, which would allow the estimation of original features $a*$, which could be later passed to inversion model. Additionally, the description also implies the discriminator that is used to train the angle-prediction. In the implementation part of section 5, authors state that the inversion model is based on the U-Net, however, there was no architecture description of the angle-estimator (which we assumed is an another neural network), nor discriminator that is used for training of such estimator. What obfuscates it further is the fact that the authors did not make clear distinctions between the aforementioned attacks (1) and (2) in the "implementation details" section, confusing as to which attack and in what way does the aforementioned U-Net architecture actually relate.

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
