# OpenReview forum: "Reproducibility report for "Interpretable Complex-Valued Neural Networks for Privacy Protection""
_ML_Reproducibility_Challenge/2020 — RC2020_

### Official Review · AnonReviewer2 · 2021-03-01
**Very good points to question but needs more evidence**

**Rating:** 9
**Confidence:** 3

**Review:**

Thank you for your great paper!

Summary: The authors tried to reproduce the original paper which claimed that complex-valued DNNs effectively increase the difficulty of inferring inputs for the adversary attacks compared to the baseline and the proposed privacy-protecting complex-valued DNN effectively preserves the accuracy when compared to the baseline, but did not get the satisfying results. Moreover, the authors proposed certain good points to question about the original paper.

- Pros:
1. The authors are really rigorous to provide certain good points to doubt the original paper (Section 5 Discussion in this paper). It proves that the authors read the papers carefully and devotes themselves to designing the experiments.
2. The authors describe the details of the experiment very carefully, and the idea is clear to me.

- Cons:
1. The authors haven't carried out the whole experiments (such as the authors did not successfully implement VGG-16 / Alexnet and the inference attacks)
2. The result section is confusing to me as I can't figure out which one is the new results without ReLU in the generator

Overall, I really appreciate the discussion section, so I would clearly recommend the paper to be accepted!


**Familiar With The Original Paper:**

I have not read the original paper

**Reproducibility Summary:**

Report has summary

---

### Official Review · AnonReviewer3 · 2021-03-08
**A good work**

**Rating:** 7
**Confidence:** 3

**Review:**

The authors did well by reproducing the original work even though there was no readily available data.

**Familiar With The Original Paper:**

I have not read the original paper

**Reproducibility Summary:**

Report has summary

---

### Decision · Program_Chairs · 2021-03-31

**Decision:**

Accept

**Comment:**

Very well-written report that goes above and beyond the reproducibility aspect and offers important insights on the original paper.